# OpenReview forum: "Tuning Language Models by Proxy"
_colmweb.org/COLM/2024/Conference — COLM_

### Official Review · Reviewer_DYpF · 2024-05-03

**Rating:** 8
**Confidence:** 4
**Ethics Flag:** 1

**Summary:**

This paper introduces proxy-tuning, a resource-efficient method that adapts LLMs without needing access to their parameters. By tuning a smaller LM and applying the predictive differences to a larger, untuned model, proxy-tuning effectively mimics full-scale tuning. Experiments show it can nearly match the performance of fully-tuned models in various benchmarks, including knowledge, reasoning, and safety, and even surpasses them in truthfulness on specific tests. The technique is also applicable to domain-specific adaptations and updating models with recent knowledge.

**Questions To Authors:**

1. Why does the proposed method adjust the output probability of the larger model using the difference between the tuned small model and the untuned small model? Why not directly adjust the probability of the larger model based on the tuned model?
2. Does the proposed model adjust the output probability of the larger model step by step, or does it adjust only once after generating all content?
3. If closed-source LLMs do not provide output probabilities or logits, does it mean that the proposed method will not work?
4. In Table 1, could you provide examples of tuned LLAMA2-13B and compare them with cases of proxy-tuned LLAMA2-13B?
5. Could you clarify why, in Table 2, Directly Tuned 70B underperforms Directly Tuned 13B? Additionally, why does Proxy-Tuned 70B significantly outperform Directly Tuned 70B in the last column, yet underperforms Directly Tuned 70B in other cases?

**Reasons To Accept:**

1. Research on how to customize and efficiently "fine-tune" LLMs is important.
2. The proposed proxy-tuning is innovative.
3. The paper is well-written, and its experiments are solid.

**Reasons To Reject:**

1. There have been some similar ideas to this paper’s idea, such as DEXPERT for detoxification and VDD [2] for reducing hallucinations or bias. It's unclear how much we can benefit from this paper’s idea.

[1] Debiasing Multimodal Large Language Models. https://arxiv.org/pdf/2403.05262

---

> ### Author Rebuttal · Authors · 2024-05-31
>
> Thank you for your thoughtful questions, which we address below.
>
> **Discussion of past work**
>
> DExperts and VDD (and many other methods) use logit arithmetic to steer the generation in a desirable way, whereas we show that it is possible to **achieve the effect of *finetuning* at decoding-time**. We believe that this is a surprising finding.
>
> Moreover, we have found that proxy-tuning is equivalent to tuning $\mathcal M$ with the implicit reward model underlying tuning of $\mathcal M^-$ (see response to w3N3)! Thus, proxy-tuning has some special theoretical properties.
>
> **Why use the anti-expert?**
>
> Following your suggestion, we experiment with an ablation without the anti-expert, which we will add to the next revision of the paper.
> Specifically, we additively combine $\mathcal M$ and $\mathcal M^+$ with a hyperparameter $\alpha$ applied to $\mathcal M^+$. We use 200 examples sampled from the full test set. We find that **proxy-tuning consistently outperforms the ablation without the anti-expert**. For AlpacaFarm, the best setting $\alpha=1.0$ gives a win rate of 81.5% compared to 90% for proxy-tuning; for GSM, the best setting $\alpha=0.4$ gives 26% accuracy compared to 26.4%. Note that $\alpha$ is very task-sensitive!
>
> We conclude that proxy-tuning works out-of-the-box, while ablating the anti-expert can give strong performance with a task-sensitive hyperparameter search.
>
> **Step by step?**
>
> Yes, we adjust the output logits at every time step.
>
> **If LMs do not provide output probabilities, does the method work?**
>
> Our method does require access to output probabilities or logits. However, a recent paper ([Carlini et al., 2024](https://arxiv.org/abs/2403.06634)) showed that even when these logits are not provided, they can be reverse engineered — in fact, the authors recovered the entire logit distribution from GPT-3.5-turbo using many queries! Moreover, proxy-tuning may be applicable even with partial access to output logits, as shown in §7.
>
> **Example generations from Llama2-13B-chat**
>
> See https://postimg.cc/FdrGs1Jx.
>
> **Discussion of TruthfulQA findings**
>
> For TruthfulQA in the open-ended setting, there are two interesting findings:
> 1. The 13B-chat slightly outperforms 70B-chat (by 0.8%). We suspect that the benchmark is saturated with scale.
> 2. The proxy-tuned models outperforms their directly-tuned counterparts. As TruthfulQA is a knowledge-intensive task, we hypothesize that decoding-time algorithms can better preserve knowledge than direct tuning.

---

> > ### Comment · Reviewer_DYpF · 2024-06-07
> > **Response to Authors**
> >
> > Thanks for your detailed clarification. Although my concerns have been addressed, I will not increase my score as I have already assigned a very positive one. Best of luck!

---

### Official Review · Reviewer_wuw1 · 2024-05-03

**Rating:** 7
**Confidence:** 3
**Ethics Flag:** 1

**Summary:**

The paper introduces a technique reminiscent of contrastive decoding to 'tune' a large LLM without gradients by shifting the logits of each next-token distribution using the difference between logits under two smaller models (one which is tuned for the relevant task and another which is not).

Given iterative access to all relevant generators and next-token distributions, the technique is fairly straightforward to apply and the paper demonstrates its effectiveness across various benchmarks. Most experiments adapt Llama 13b or 70b using adapted and unadapted versions of Llama 7b. In one experiment, the paper demonstrates (in a rather limited scenario, where a single-token response is sufficient) that the technique can be used in a setting as closed-access as that of adapting ChatGPT (provided access to top-5 logits).

The paper is reasonably clear, and complete (to the best of my ability to assess it).

**Questions To Authors:**

In one experiment reported in this paper, proxy tuning beats direct tuning in Truthful QA, this can be due to anything, something about the method, something about the data, something about this one experiment.

I think the paper would be better without the following attempt at an "explanation" _"The improvement in truthfulness suggests that decoding-time algorithms may preserve knowledge better than direct finetuning, which sometimes hurts knowledge-intensive tasks"_. Besides being shallow, it doesn't explain, it makes further claims that you cannot test easily.

**Reasons To Accept:**

A clearly written paper that presents a simple and effective decoding-time technique to adapt a large base LM without access to its internals (but requiring iterative access to its generation algorithm and requiring the logits that parameterise the next-token distribution at each step).

**Reasons To Reject:**

I can list a point for improvement, but I think it only requires clarification before a final revision (ie, it's not a reason to reject the current version).

It's not too clear to me when this method would be needed. When do we have access to all that it requires but do not have the ability to tune the large model? The example of ChatGPT isn't a good one: while it allows a demonstration of the method, that demonstration is in a rather artificial and limiting single-token response setting.

I think it's okay if the answer to this question can be seen as a bit disappointing, but I think this should be discussed clearly and openly in the paper.

---

> ### Author Rebuttal · Authors · 2024-05-31
>
> Thank you for your thoughtful suggestions! We are grateful that you recognize the simplicity and effectiveness of the method. We hope that the following discussion addresses your concerns.
>
> **When would this method be needed?**
>
> You are definitely right that currently, there are not many situations where model producers provide logit distributions but not model parameters.
>
> Our response is three-fold:
> 1. While proxy-tuning enables tuning of black box models (and this is our main motivation), **it has many benefits even for white box models**. For example, it has been useful for a distributed learning setup where the expert is tuned on data that must stay on-device, and therefore cannot be used to tune the larger base model. Proxy-tuning also enables a “tune once, proxy-tune many” setup, where hundreds of models can be improved for the training cost of tuning one model. Even with access to parameters of the base model, tuning extremely large models requires much more resources than proxy-tuning with small (anti-)experts (we used TPUs to finetune the 70B model, while three A100 GPUs is enough to proxy-tune the 70B model). Finally, the strength of tuning is controllable in proxy-tuning (as we show in §6.2), which is not true for direct tuning.
> 2. Future work may be able to **indirectly obtain model logits**. For instance, a recent paper ([Carlini et al., 2024](https://arxiv.org/abs/2403.06634)) reconstructed the *entire* logit distribution from GPT-3.5-turbo through multiple queries (something we did not know!). Although OpenAI changed the functionality of the API in response, the paper shows that it is still possible to recover the complete logit vector at an increased cost.
> 3. Very wishfully, we hope that proxy-tuning incentivizes model producers to provide logits in the future, because of the greater user customization it enables. Thus, even though the current setting is not very common now, perhaps it will be more common in the future!
> Following your suggestion, we will provide more discussion of this in the paper.
>
> **The paper would be better without the "explanation" for TruthfulQA results**
>
> We really appreciate your suggestion. We definitely intend for it to be a hypothesis, not an explanation. We will rephrase it to say “*Direct tuning has been shown to sometimes hurt performance on knowledge-intensive tasks, and it is possible that decoding-time algorithms provide an avenue for better knowledge preservation.*”

---

### Official Review · Reviewer_cyoK · 2024-05-12

**Rating:** 8
**Confidence:** 4
**Ethics Flag:** 1

**Summary:**

This paper tackles a problem setup that, given a small accessible LM and a large inaccessible LM, how we can close the performance gap between the small and large model. This setup is highly relevant to many researchers and practitioners who lack the resources to train the largest LMs, such as those with 70 billion parameters.

The proposed approach is fairly simple: First, it fine-tunes a base model on the target task, then adds the logit offsets between the tuned and base models to the logits of the larger (untuned) model, followed by softmax. The adjusted distribution is then used for the final generation by the large model.

The authors rigorously evaluate this approach on a range of target tasks, from instruction tuning to code adaptation. The results of instruction-tuning experiments strongly support the effectiveness of this approach, achieving the win rate of 88% on AlpacaFarm and 32% on GSM. Proxy tuning even outperforms the directly tuned model in terms of accuracy on TruthfulQA, which intensively requires world knowledge. These positive trends are consistent in the code adaptation experiments as well. Additionally, the authors demonstrate a use case of proxy tuning with proprietary LMs such as GPT-3.5, which boosts the performance on RealTimeQA.

**Questions To Authors:**

- Have you observed any failure modes after applying proxy tuning? Particularly, it would be interesting to see losses where the base model gets right, but proxy tuning gets wrong.
- Have you investigated different temperature values in softmax? Or, a vanilla softmax just works fine? I wonder if this is task dependent.

**Reasons To Accept:**

- A simple yet effective approach for managing large language models (e.g., 70B-parameter models) in limited-resource settings. This approach opens up opportunities to researchers and practitioners who lack the resources to train the largest LMs.
- The strong experimental results demonstrate the effectiveness of this approach.
- This paper is very well-written, clearly describing the approach and its evaluation.

**Reasons To Reject:**

- Although this approach doesn’t require updating a large LM itself, it still requires training a small LM and performing forward computation for three times during inference time. This might be costly in practice.
- This approach assumes a situation where a small accessible LM and a large inaccessible LM are available. Therefore, this approach can be used for LLM families that release different sizes and versions. Although this is typically not a problem in practice (i.e., model developers usually release different sizes and versions), one could argue that it is a strong assumption.

---

> ### Author Rebuttal · Authors · 2024-05-31
>
> Thank you for your insightful questions, and we are grateful that you recognize the effectiveness of the approach.
>
> **Inference cost**
>
> Proxy-tuning does incur a greater inference-time cost, which we quantify in §C.1. We note that the increased runtime is due to a sequential execution of the models in proxy-tuning; in practice it can be greatly accelerated by deploying on multiple GPUs in parallel that communicate with each other.
>
> In addition, as models push the limits of scale and available training data, we believe inference-time methods will be important to pushing model capabilities further. In many cases, users may be willing to wait longer for a better generation.
>
> **Assumes a situation with small accessible LM and large inaccessible LM**
>
> Note that the small and large pretrained models do not need to be in the same model family, as long as they share the same vocabulary (or at least enough overlap for the task of interest), as we showed for applying Llama-2 models to steer GPT-3.5. Excitingly, very recent work ([Minixhofer et al., 2024](https://arxiv.org/abs/2405.07883)) developed a method that swaps a LM’s tokenizer with an arbitrary new one, and should be able to **alleviate the requirement of shared tokenizers in proxy-tuning**!
>
> Moreover, while proxy-tuning enables tuning of black box models (and this is our main motivation), **it has use cases outside of black box settings**. For example, it has been useful for a distributed learning setup where the expert is tuned on data that must stay on-device, and therefore cannot be used to tune the larger base model. Proxy-tuning also enables a “tune once, proxy-tune many times” setup, where hundreds of models can be improved for the training cost of tuning one model. Even with access to parameters of the base model, tuning extremely large models requires much more resources than proxy-tuning. Finally, the strength of tuning becomes controllable for different use cases (as we show in §6.2), which is not true for direct tuning.
>
> **Failure modes**
>
> We did not observe any consistent failure modes.
>
> **Investigation of different temperature values**
>
> We use temperature in the code adaptation experiments (§4), following the same decoding hyperparameters as the Codex paper ([Chen et al., 2021](https://arxiv.org/abs/2107.03374)) with temperature = 0.8 and top $p$ = 0.95. For a consistent comparison, we did not explore other options. These hyperparameters are meant to encourage diversity in sampling.

---

### Official Review · Reviewer_w3N3 · 2024-05-22

**Rating:** 7
**Confidence:** 3
**Ethics Flag:** 1

**Summary:**

This paper proposes the proxy tuning method, which uses tuned and untuned small language models to steer the predicted logits of a larger LM at decoding time. The predicted logits are offset with the difference between the logits of tuned and untuned small LMs. The authors evaluate their proposed method of instruction-tuning, domain adaption and task-specific for the LLaMA-2 family and temporal adaptation for the proprietary model (GPT-3.5). The experiment results show that the proxy-tuned LMs significantly outperform the base models and approach the performance of the directly tuned models in some cases.

**Questions To Authors:**

1. How do the predicted logits of proxy-tuned and directly-tuned models look like? Do the proxy-tuned logits have a similar pattern to the directly tuned model?

**Reasons To Accept:**

- While the proposed method is simple, it is efficient and works well in multiple scenarios.
- Well-designed experiments to show the benefits of the methods in common LLM usecases.

**Reasons To Reject:**

- It is intriguing to understand the underlying conditions and reasons for the method's effectiveness. The authors' analysis or hypothesis on this matter would undoubtedly enrich the paper. I speculate that, in addition to requiring the same vocabulary, it is crucial for the small and large LMs to be trained to approximate the same data distribution. This could explain why the proxy-tuned model cannot surpass the CodeLlama-7B directly-tuned model, as CodeLlama-7B has been adapted to a new data distribution (code data). This can also explain the limitations in the minor improvement of GPT-3.5 experiment.

---

> ### Author Rebuttal · Authors · 2024-05-31
>
> Thank you for your insightful comments, and we are grateful that you recognize the effectiveness of the method in many use cases.
>
> **Underlying reasons for effectiveness**
>
> Since submission, we have developed a better theoretical understanding of proxy-tuning, which we summarize below.
>
> Given a pretrained model $\mathcal M$, the objective for RL with a KL divergence penalty (used by e.g., PPO) is defined by
>
> $$\operatorname{argmax}\_{\mathcal{M}^\ast} \underset{y\sim P\_{\mathcal{M}(\cdot\mid x)}}{\mathbb{E}} r(x,y) - \beta \operatorname{KL}( P\_{\mathcal{M}}(\cdot \mid x) \mid\mid P\_{\mathcal{M}^\ast}(\cdot \mid x))\quad\quad\quad\text{(1)}$$
>
> This is well-known to have a closed form solution ([Korbak et al., 2022](https://aclanthology.org/2022.findings-emnlp.77/)),
>
> $$P_{M^*}(y \mid x) = \frac{1}{Z} P_{M}(y \mid x) \exp \left(\frac{1}{\beta} r(x,y)\right)\quad\quad\quad\text{(2)}$$
>
> More generally, this means *any* finetuned model $\mathcal M^\ast$ can be viewed as implicitly optimizing an underlying reward $r$ given by
>
> $$r(x,y)=\beta \log\frac{P_{\mathcal M^*} (y\mid x)}{P_{\mathcal M} (y\mid x)},\quad\quad\quad\text{(3)}$$
>
> as can be seen by substituting Eq (3) into the RHS of Eq. (2) to recover the LHS.
>
> Now, proxy tuning is
>
> $$P_{\mathcal M^*}(y \mid x) \propto P_{\mathcal M}(y \mid x) \frac{P_{\mathcal M^+}(y \mid x)}{P_{\mathcal M^-}(y \mid x)}\quad\quad\quad\text{(4)}$$
>
> where $\mathcal M^+$ and $\mathcal M^-$ are small tuned and untuned proxies. Thus **proxy-tuning can be seen as tuning $\mathcal M$ with the underlying reward for tuning of the small anti-expert**, $r(x,y)=\beta \log\frac{P_{\mathcal M^+} (y\mid x)}{P_{\mathcal M^-} (y\mid x)}$. Intuitively, _to the extent_ that the tuning of $\mathcal M$ and of $\mathcal M^-$ correspond to the same underlying reward function, proxy-tuning is equivalent to true finetuning.
>
> **Analysis of logits from proxy-tuned and directly-tuned models**
>
> Following your suggestion, we compare the proxy-tuned and directly-tuned models in terms of the KL div in their predictions, when conditioning on the same prefix. We used AlpacaFarm prompts and only looked at the prediction for the first time step of generation. We find that the median KL div between the proxy-tuned and directly-tuned models is 0.147, compared to 0.239 between the base (untuned) model and the directly-tuned model. This means that **the proxy-tuned probability distributions are indeed more similar to those of the directly-tuned model**!

---

> > ### Comment · Reviewer_w3N3 · 2024-06-06
> >
> > The authors have addressed my concerns properly and would stay positive towards it.

---

### Decision · Program_Chairs · 2024-07-10

**Decision:**

Accept

**Comment:**

This work proposes proxy-tuning, a method based on the assumption that you have a large unaccessible LLM and a small accessible one, and that train large LLM once proxy tune small LLM many times makes sense. The proposed idea is a decoding time  algorithm that operates on top of the large LLM to achieve the same goal as direct finetuning.

pros

- The proposed idea is useful when the assumptions hold in practice, which is some of the time such as finetune on device.
- the paper is clearly written
- the experiments results are interesting
- adding the theoretical results that authors mentioned during rebuttal is helpful for the paper.

cons
- I suggest not using a single dataset such as Truthfull dataset and generalizing that the proxytuned models are more truthful.  given the nature of LLMs we can only say for this specific case unless working on a variety of datasets.
- adding clear details of scenarios where this proposal is useful in practice and discussion of costs is important. The comment from authors that sometimes folks are ok with waiting longer to get better results, is not a solid justification